# Enhancing Occupants' Thermal Comfort in Buildings by Applying Solar-Powered Techniques

Abdul Munaf Mohamed Irfeey [1,*], Elmira Jamei [2], Hing-Wah Chau [2] and Brindha Ramasubramanian [3]

1   Department of Biosystems Technology, Faculty of Technology, South Eastern University of Sri Lanka, University Park, Oluvil 32360, Sri Lanka
2   Institute for Sustainable Industries & Liveable Cities (ISILC), Victoria University, Melbourne, VIC 3011, Australia
3   Center for Nanotechnology & Sustainability, Department of Mechanical Engineering, College of Design and Engineering, National University of Singapore, Singapore 117575, Singapore
*   Correspondence: ifrmunaf@gmail.com

**Abstract:** As most people spend their days indoors, it is indeed important that buildings provide residents with a higher standard of health, convenience, and safety. As a result, many practices are implemented into buildings to improve the comfort of occupants, particularly thermal comfort; nevertheless, the energy required to run and maintain these applications is a significant constraint. Renewable energy sources offer alternative solutions to energy demand problems, and selecting the best renewable energy sources is crucial. In this article, we examine the health and well-being advantages to the occupants, as well as the surrounding environment, of a variety of novel strategies that may be integrated into buildings to increase occupants' thermal comfort for conventional practices using solar power. The key discoveries explored in this article include daylighting, passive ventilation, thermal applications, cooling applications, and power generation. For this, the information was gathered by a systematic review of the relevant prior literature. In addition, the detrimental effects of existing practices on the health and well-being of residents and the environment are included. While there are still some practical obstacles to overcome in the extraction of solar energy, the technology exists. Potential future obstacles to the broad acceptance and usage of solar energy systems in buildings are highlighted, as well as possible solutions.

**Keywords:** occupant health and well-being; energy efficiency; indoor environmental quality; daylighting; passive ventilation; solar thermal applications; solar cooling applications





## 1. Introduction

Rapid population increase has necessitated the formation of a harmonious relationship between humans and the dynamic natural environment; exposure to its rich and ever-changing stimuli has played a significant part in establishing individual life patterns, raising the pace of industrialization and laying the way for increasingly denser, building-dominated urban areas. The United Nations projects that by 2050, the global population will reach 9.7 billion, and that 1.6 billion new buildings will be constructed [1]. According to data from the World Bank, there were about 794,432 built-up areas (BUAs) globally in 2016. This number is predicted to rise to 849,407 by the end of 2022 [2]. As the majority of the average individual's daily activities take place inside the built environment, which includes residential, industrial, commercial, and recreational facilities, it is crucial that the state of the buildings, which should be more pleasant and healthier for the tenants, be maintained in order to promote a healthier and more productive lifestyle.

Occupants' comfort and well-being are ensured by thermal comfort, visual comfort, acoustic comfort, and indoor air quality [3]. Thermal comfort, among other things, is crucial in buildings in improving the quality of life of its occupants. Failing to adopt a good

practice for these, ultimately, creates discomfort for the occupants and results in building-related sicknesses, called sick building syndrome (SBS), such as asthma, hypersensitivity pneumonitis, and respiratory infections such as Legionnaires' disease [4]. In addition to these, mucous membrane and skin irritation, central nervous system problems such as headache, weariness, trouble concentrating, and chest tightness; and dislike of scents or tastes are also very serious issues [5]. Therefore, implementing certain utility measures, including heating, ventilation, and air conditioning (HVAC), lighting, and soundproofing structures, with fit-to-purpose ergonomics will improve the quality of life of occupants.

Hence, it is difficult to find the energy sources necessary to power the applications that improve the comfort of building occupants. Generally, buildings account for the majority of overall energy use in urban areas. They use around 42% of the global annual energy supply [6]. This energy is often used to provide thermal comfort, refrigeration, lighting, communication and entertainment, sanitation and hygiene, and food, among other comforts. Despite this, the availability and accessibility of the necessary energy is a limiting factor, due to the fact that the majority of energy sources are fossil fuels, which are quickly dwindling. In addition, fossil fuels are by far the main cause of global climate change, accounting for over 75% of global greenhouse gas emissions and almost 90% of carbon dioxide emissions [7]. As a result of shifting precipitation patterns, climate extremes, and the effects of air pollution, climate change may endanger the safety and well-being of building occupants through increased temperatures, more frequent heat waves, the occurrence of heat-related illness, and higher mortality rates [8].

Further, the majority of older types of air conditioners rely on CFCs and HFCs, which contribute significantly to global warming. Even modern types, which depend more on HFCs and HFOs, are a significant contributor to ozone depletion. In addition, the major issue with incandescent lights is that they waste a great deal of energy. In order to generate heat, the light bulbs need an enormous quantity of power, since it requires a significant amount of energy for the bulbs to warm the filament. Over the life cycle of a building, the HVAC systems that provide comfortable conditions throughout the utilization phase absorb 94.4% of the total energy consumed [9]. To decrease this rate, passive approaches and renewable energy sources should be employed to give comfort, rather than mechanical systems. In this way, buildings may be constructed with physical conditions more conducive to human health.

Focusing on this, energy-efficient renewable energy technologies that are less detrimental to the surrounding environment and have fewer negative effects on building occupants are the alternative solution for these issues. Globally, solar energy is the most plentiful renewable energy source, and thermal and photovoltaic energy forms might potentially provide a solution and greatly decrease dependency on fossil fuels. Adopting solar energy harvesting technologies in buildings will be a potential source of energy to operate and maintain regular utility practices, including HVAC and lighting. Consequently, the aims of this study are to investigate the ways of utilizing solar energy for various purposes, and to examine the possibilities of these practices to enhance the comfort and well-being of the occupants in the building.

The specific objectives of the conducted study can be summarized as:

- Determining the factors influencing building-related health problems and discomforts;
- Identifying numerous amenities to increase building occupant thermal comfort;
- Analyzing methods to improve occupant thermal comfort in buildings;
- Assessing current practices' negative effects and innovative methods' potential health and well-being advantages.

The structure of this study is provided below. Section 1 includes the introduction and the scope of the work. Section 2 describes how this research evaluated and analyzed the different objectives and data gathered from the literature. Section 3 examines in detail building-related discomfort and diseases. Sections 4 and 5 address the subject of occupant comfort enhancement using solar energy, and discussion respectively. Sections 6 and 7 include the future scope and conclusions.

## 2. Literature Studies

The thermal comfort of occupants is significantly associated with the indoor air temperature, lighting, ventilation, and relative humidity of the buildings [10]. These variables hinder the occupants' health and well-being, and the ability to focus on their work. Generally, allergy and immunologic problems, infections, and exposure to chemicals and other pollutants are the root causes of building-related illnesses [5]. Chronic fatigue syndrome (CFS) is characterized by acute and persistent tiredness that impairs daily activities [11]. Pesticides, organophosphates, solvents, and other substances, in addition to a variety of psychological, environmental, and behavioural variables, have been linked to the development of CFS.

The age of the buildings, inadequate ventilation, broken windows, and inefficient HVAC systems have all been implicated in these illnesses. Contrary to common opinions, the impact of heat on human performance is complicated and dependent on elements other than temperature alone, including radiation, wind, and humidity. Owing to its evident effect on human health, humidity has gained a disproportionate amount of attention among these factors. In the presence of humidity, microorganisms such as mildew and mould have a strong chance of proliferating, producing a potentially harmful scenario for the occupants' health. Symptoms of excessive humidity include muscle cramps, disorientation, and heat stroke; it may also exacerbate previous conditions such as asthma and heart disease [12].

Indoor air pollutants include oxides of nitrogen ($NO_x$), carbon monoxide (CO), carbon dioxide ($CO_2$), volatile organic compounds (VOCs), and particles, which are released by utility applications such as HVAC, lighting, and maintenance operations [13]. As a consequence of a rise in $CO_2$, ailments such as sore throat, nasal irritation, mucous membrane symptoms, headaches, and tight chest are more common among building residents [14]. When volatile organic compounds (VOCs) such as o-xylene, styrene, d-limonene, and other terpene compounds mix with ozone and $NO_x$ entrained from the environment, aldehydes and ultrafine particles may be produced [15].

As a result of exposure to air pollution and other environmental hazards, oxidative stress may contribute to the development of SBS. Many air contaminants include reactive oxygen species (ROS), which may cause oxidative damage to lipids, proteins, and nucleic acids [16]. When temperatures are kept above the thermal comfort zone of the inhabitants, it causes a rise in symptoms such as headaches, weariness, and mucosal irritation [17]. Regardless of the efficacy of ventilation systems, the number of electronic devices utilized in the workplace increases the heat loads of buildings. In colder regions during the summer, humidification will alleviate SBS symptoms and the sense of dry air by adding moisture [18]. Because relative humidity (RH) in a tropical setting stays much greater, even in an indoor workplace, it is likely that air dryness or humidification are not the cause of SBS in such circumstances.

Owing to exposure to moisture and bio-aerosols, buildings often emit musty or otherwise unpleasant scents, which are best understood in the context of disease through physiological symptoms. SBS symptoms include decreased productivity, nose and throat irritation, headaches, tiredness, asthma, rhinitis, and an increased susceptibility to colds and influenza. Insufficient airflow in confined settings diminishes vigilance, attention, memory, and concentration. SBS, including inflammation, respiratory infections, asthma symptoms, and short-term sick leave, are associated with higher ventilation rates inside buildings, up to around 25 L/s per person [19]. Inconveniences resulting from inadequate illumination may pose a risk to people's health. The suprachiasmatic nucleus (SCN), a region of the hypothalamus important for controlling circadian rhythms, is regulated by light through the eye and the retinohypothalamic tract. Furthermore, exposure to light decreases melatonin synthesis, which might result in overnight awakenings [20].

## 3. Methods

As a consequence of a scarcity of land space and the concentration of a massive population inside metropolitan borders, the expanding requirements of the human popu-

lation have necessitated the construction of denser structures. The majority of everyday activities are conducted inside the built environment for a variety of reasons; thus, the building should be able to create a pleasant and healthy setting in which to spend time and accomplish the objectives effectively.

Figure 1 describes the procedures to conduct this study. The whole study adhered to the PRISMA-P protocols, in order to perform a systematic evaluation of previously-published studies on thermal comfort-enhancing applications and their adverse effects on environmental and human health and well-being. The use of solar energy with different technologies to extract energy in order to run utility practices to enhance the interior and exterior thermal comfort of buildings, as well as the health benefits for the occupants, is explored. The purpose of the research was to evaluate inventive revolutionary ideas that may be used in both new and existing structures. The provision of certain features within a building is crucial to ensure the comfort of its occupants. These features include ventilation, heating, cooling, lighting, and access to electricity for additional amenities. All of these aspects were considered in the present study. For the study, the majority of literature papers published between 2012 and 2023 were evaluated. This study was undertaken in four phases. The objective of this study's stage 1 was to conduct a comprehensive online literature review, to assess the discomfort and diseases linked with buildings. In addition to articles from scientific databases (Science Direct, Scopus, and Google Scholar), the online literature includes books, university reports, and periodicals. The review's data were maintained in an electronic database for easy access. The collected prior research was assessed to determine relevance, eligibility, and inclusion within the scope of the present investigation. Books, university reports, and university publications provided us with access to university materials. This content was required to be as updated as feasible, and to include all relevant information. Some of the available data lacked the required degree of timeliness. In the literature review, more than 200 publications were reviewed, but 98 were inappropriate to the study's objectives and out-of-date; hence, they were excluded.

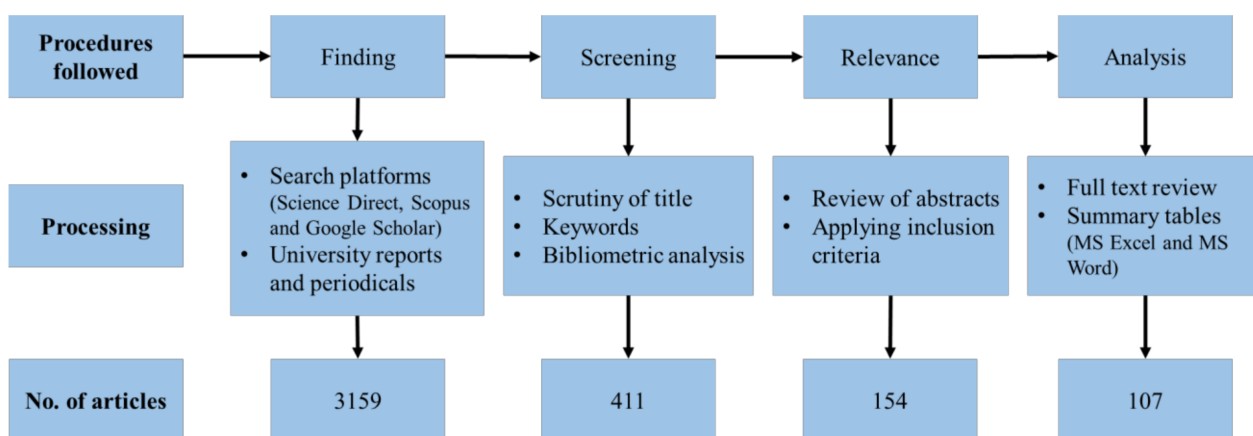

**Figure 1.** Procedure for conducting this study.

In stage 2 of this article, methods for extracting solar energy as an alternative to current practices were identified. In addition, the means by which such technologies improve the health and well-being of the tenants, and the negative effects of typical practices were investigated. In stage 3 of this study, discussions about the current practices and issues with the wide applications using solar energy technology in buildings were assessed in light of earlier studies. The future scope and conclusions were examined and explained in stage 4.

## 4. Strategies to Improving the Occupants' Thermal Comfort

Improvements in human health and thermal comfort within the built environment are analyzed from a variety of perspectives, since SBSs are a crucial phenomenon that

must be addressed for a more productive and healthier life inside the built environment. Effective measures should be taken to meet fundamental requirements including electricity, lighting, heating, ventilation, cooling, and indoor air quality standards [21]. The ability of buildings to maintain a steady internal temperature is crucial to fostering an atmosphere that is conducive to people's complete needs. As energy sources are the requirement for providing such thermal comfort enhancement, sustainable energy sources with minimal negative consequences to the occupants as well as environment are vital. Renewable energy solutions are becoming more popular across the globe as a method of satisfying the ever-increasing energy requirements for keeping building occupants in optimal comfort and safety.

Many variables must be taken into account when developing an intervention to improve a building's energy efficiency, including the historical elements that need to be preserved, the building's intended use, and the energy and comfort needs of the building's occupants [22]. Interventions targeted at minimizing the consumption of the building must be evaluated early on in the design process. This includes high-performance window frames, well-insulated opaque surfaces and claddings, efficient thermal systems, and passive approaches for heating and cooling [23]. In order to reduce the energy consumption of the building, options including efficient thermal systems and passive approaches for heating and cooling must be evaluated early in the design process. According to a prior research investigation, it was demonstrated that passive buildings possess the capability to conserve energy in all global climate zones, as they utilize between 75% and 95% less energy than traditional buildings [24].

Old buildings are often designed according to outdated standards that cannot provide the same levels of comfort and efficiency as newer structures [25]. More in-depth conversation between all the technical area specialists and all the concerned stakeholders is necessary in order to achieve local and international legal or regulatory requirements towards new designs, to enhance the thermal comfort of the occupants and the energy efficiency. This emphasizes the need to formulate a procedure for work that is as dispassionate as possible, so as to lessen the likelihood of inconsistencies resulting from subjective and arbitrary evaluations. Furthermore, a correct approach is required that evaluates all parameters in an integrated assessment, studying the building components as well as the typological and functional parameters in order to determine the most effective applications.

Incorporation of solar energy technologies into buildings varies by region, building type, and significance [26]. Initiatives at any level that focus on the design of energy-efficient systems may benefit both new and existing buildings. The selection of the optimal energy types of the solar energy for adaptation is hampered by issues such as the changing energy demands of different building utility applications, as well as considerations concerning knowledge, technology, economic feasibility, maintenance, and operation [27]. More disputed, however, is the topic of how to attain energy efficiency in building design, especially as to whether it is preferable to prioritize the retrofit of existing buildings, or the demolition of existing buildings and construction of new structures. Carbon emissions might be reduced more rapidly via building retrofits. Significant grounds for retrofitting include the fact that two-thirds of the global building stock will consist of existing structures and the slow pace of building destruction and reconstruction [28].

### 4.1. Passive Ventilation

The heating, ventilation, and air conditioning of a building are major contributors to the quality of the ventilation within. In order to keep interior climatic conditions comfortable, air conditioning systems may be used to chill, heat, and dehumidify or humidify air delivered to rooms. Nevertheless, HVAC systems may provide either external air, or a mixture of outdoor air and air pulled from the inside rooms to the spaces being heated or cooled. However, HVAC systems can contaminate the air supplied to the rooms in a number of ways, including the systems themselves being contaminated, which then contaminate the air that flows through them; the systems drawing contaminated air into

the building from outdoor sources; and the systems that recirculate indoor air, spreading the contamination generated at specific indoor locations throughout the building [29–31]. The development of moulds and other microbes provides a risk of HVAC system contamination, especially in areas with high humidity or liquid water, such as on or near air conditioning, dehumidification, and humidification equipment. Deposited dusts, oil residues from production, insulation, and sound-absorbing materials are all possible causes of contamination in HVAC systems.

Studies have shown that people with SBS are more likely to have symptoms in air-conditioned facilities, including nervous system symptoms, and upper respiratory and mucous membrane symptoms [32,33]. Symptoms of the skin and lower respiratory system are less strongly-linked to air conditioning. However, the use of air conditioning systems that use liquid water-based humidification are linked to an increased prevalence of some symptoms when compared to the use of air conditioning systems that did not use humidification at all [33]. It is unlikely that the kind of HVAC system would have any direct effect on health symptoms, but it might be a proxy for an exposure or exposures that do impact symptom prevalence.

Natural passive ventilation is a great strategy for cutting down on the high supplemental energy use for ventilation. Recently, however, interest in natural ventilation has soared due to the realization that it may be preferable to mechanical ventilation systems in terms of health and well-being of the occupants, energy consumption, cost, and environmental impact [34]. As a result of their higher electrical needs, mechanical ventilation systems have unfavourable energy implications. Air conditioning demands might overwhelm electrical systems in certain places. Natural ventilation, on the other hand, is predicted to reduce annual energy consumption by a certain portion for cooling and for fan power when climate and operating circumstances are favourable. A significant amount of $CO_2$ is released into the atmosphere whenever a mechanical ventilation system is used.

Therefore, natural ventilation is increasingly advocated as a technique for conserving energy and providing quality air with enough thermal comfort for the occupants of buildings. Thermal buoyancy or wind may cause natural ventilation. When wind blows against a structure, for instance, it causes the pressure to increase on the windward side and decrease on the leeward side [35]. With the pressure differential acting as a driving force, ventilation is initiated when air is drawn into the structure via the windward opening and exhausted out the leeward opening. This kind of ventilation is also known as wind-driven cross-ventilation.

Increasing buoyancy effect has been used in a variety of ways to help initiate air circulation and keep the temperature inside a structure where it should be [36]. A solar chimney is a good example of a device created to improve ventilation by maximizing solar gain and, in turn, generating a large enough temperature differential between the interior and outside of a structure to motivate enough air movement [37,38]. Figure 2 depicts a solar chimney used to produce passive ventilation and air movement from the interior of a room to the atmosphere; the principal technique underlying this method is described. During the summer, there is usually just a small temperature differential between the interior and exterior of a structure. Therefore, the ventilation in a standard chimney, which uses the stack ventilation concept, is inadequate because of the insufficient thermal forcing. On the other hand, by increasing sunlight gain using a solar chimney, a significant temperature differential may be achieved.

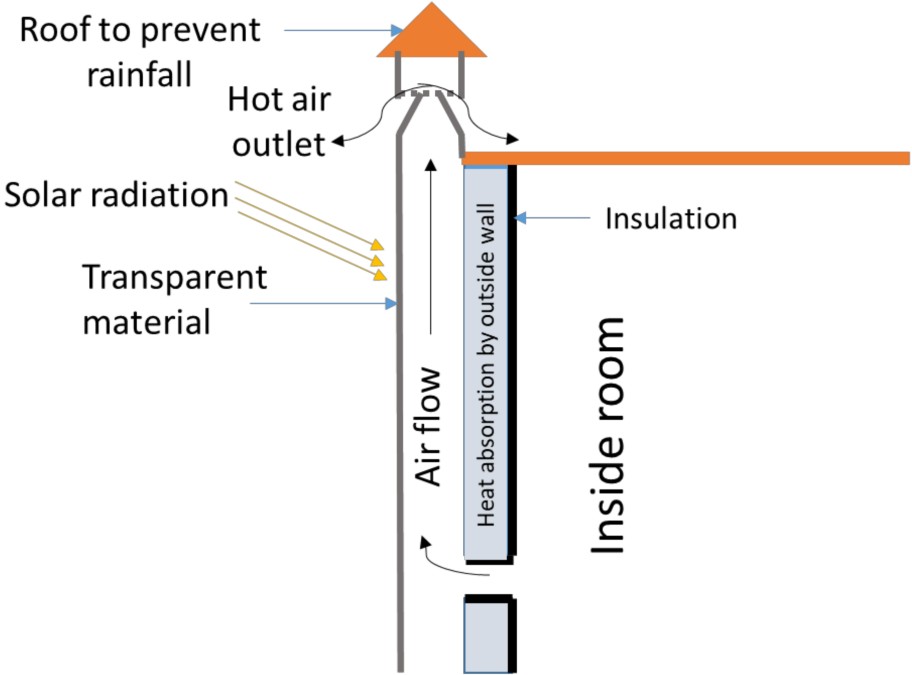

**Figure 2.** Schematic diagram of a solar chimney attached to a room.

*4.2. Solar Heating Applications*

Since hot water demands a substantial amount of energy, how to reduce the energy consumption for hot water system has to be addressed. According to estimates, water heating uses more than 30% of domestic energy usage [39]. The majority of residences and businesses obtain their energy from fossil fuels, including coal, gas, and oil. In addition to this, many individuals prefer to heat their water using electricity and firewood. Due to the environmental and health dangers posed to the surrounding environment of the built environment, water heating using fossil fuels and wood is a significant area of concern. This increases particulate matter (PM) pollution in the surrounding area, such as carcinogenic chemicals and cancer-promoting agents, and daily increases in outdoor PM concentration have been linked to increased mortality and hospitalization [40,41].

Exacerbation of respiratory disorders, such as asthma and chronic obstructive pulmonary disease (COPD), as well as bronchiolitis and otitis media, which often begin as upper respiratory infections, seems to be linked to ambient levels of particle air pollution from wood burning [42]. Ambient $PM_{2.5}$ concentrations greater than $10 \text{ g/m}^3$ increase the prevalence of respiratory, cardiovascular, and cerebrovascular illnesses, as well as premature mortality [43]. Reducing the use of fossil fuels in power plants, central heating systems, and industry is ongoing, in an effort to reduce annual average ambient $PM_{2.5}$ levels by 25% [44].

Unlike fossil fuels and firewood, solar thermal energy produces no carbon emission. In addition, the method generates no waste, causes no pollution, and has no negative environmental consequence. Solar water heaters not only cut power costs, but also provide a number of additional benefits. Active solar energy systems are an alternative method of providing space heating for a building and hot water supply. In an active system, solar collectors mounted on roofs capture sunlight and are used in conjunction with pumps and fans to recirculate the warm air. To economize on construction supplies and labour for installing the solar collectors, it is best to incorporate them as seamlessly as possible into the structural elements of new buildings. The cost savings from mass-producing the solar collectors should encourage their prefabrication as part of the building's walls or roof [45].

The position of the tank is a challenge for the system's integration into the building. It is efficient to place the tank next to the collectors [46]. In warmer areas, the thermosiphon system is often employed, in which the tank is linked immediately above the collectors. Just as with any other solar system, keeping shadows at bay is crucial. Roofs on modest high-rise buildings are preferable to walls in this regard. A higher angle will reduce the seasonal changes over the year, and a larger collector will be required if flat plate collectors are used.

The optical concentration of light is one of the solar thermal energy-harvesting methods that may be used in buildings. The amount of energy harvested from the sun may be increased by focusing more light onto a solar absorber via the use of mirrors. Two-dimensional troughs of parabolas, often oriented east–west, and three-dimensional sun-tracking dishes, typically used in conjunction with a conventional steam turbine, are common in solar energy applications [47,48].

### 4.3. Solar Cooling System

Solar cooling systems are briefly divided into two types: solar thermal cooling systems and solar electric cooling systems. In contrast to thermal systems, solar electric cooling systems use the electricity generated by solar photovoltaic (PV) panels to power a standard electric vapor compressor air conditioner, while solar thermal cooling systems use the heat from the sun to power thermal cooling systems, such as absorption, adsorption, and desiccant cycles. With the usage of any form of solar cooling, up to 95% less power is required to cool and refrigerate buildings [49]. The solar cooling system is made up of the solar energy conversion system, the refrigeration system, and the cooling load. The cooling method most suited to a given application is influenced by a number of factors, including the amount of cooling required, how it is distributed over time, the chilled object's temperature, and the heat source [50]. The temperature of the solar working fluid may be adjusted to meet a variety of needs, depending on the solar collectors used and the available sunlight. This temperature range is adaptable to different cycle needs, as to whether it requires a rather high driving temperature to be powered, or if a lower temperature is sufficient for operation.

In the solar-powered refrigeration system, the absorption refrigeration cycle is one of the applications used widely. Absorption systems often use evacuated tube collectors (ETC) since this technology can maintain temperatures at or above 80 °C [51,52]. Nonetheless, solar-concentrating collectors may be used to boost overall system performance. These kinds of cooling systems are useful in buildings where there is a large demand for electrical power to run the air conditioner. Increasing numbers of buildings are installing such systems annually. As the name implies, solar cooling systems are most common in hot and balmy climates. Fan-coil systems are used to chill the building's interior. Solar-powered adsorption refrigeration systems work by cycling between two absorbers, each of which receives energy from the secondary fluid in turn [53]. To provide cooling power, the bed's discharged refrigerant is routed via a condenser, expansion valve, and evaporator. The cooling tower receives heat from the condenser. The temperature of the solar working fluid needed to thermally drive the adsorption cycle varies with the kind of working pairs (solid and fluid). Flat-plate solar collectors that operate at temperature of 80–90 °C are viable options in regions with strong sun irradiation [54].

Solar solid desiccant is another method of solar cooling, and it consists of two air ducts, two fans, a heat wheel, a desiccant wheel, two evaporative coolers (humidifiers), and a heater [55]. The theory of operation is based on the fact that heat is removed from the incoming supply of fresh air in order to preheat the exhaust air as the building is ventilated [56]. The humidification process follows the rotary heat exchanger, which cools the incoming air. Humidification allows air to be cooled to increasingly lower temperatures with increasingly drier air entering the system. Most of the time, desiccant-based evaporative cooling (DEC) systems are used in regions with hot and humid climates. When compared to adsorption and, particularly, absorption methods, their cooling capability is

often substantially lower. As depicted in Figure 3, using solar thermal energy through solar collectors and solar photovoltaics through solar modules, two distinct types of cooling systems may be implemented. In the solar-powered refrigeration system, the absorption refrigeration cycle is one of the applications used widely.

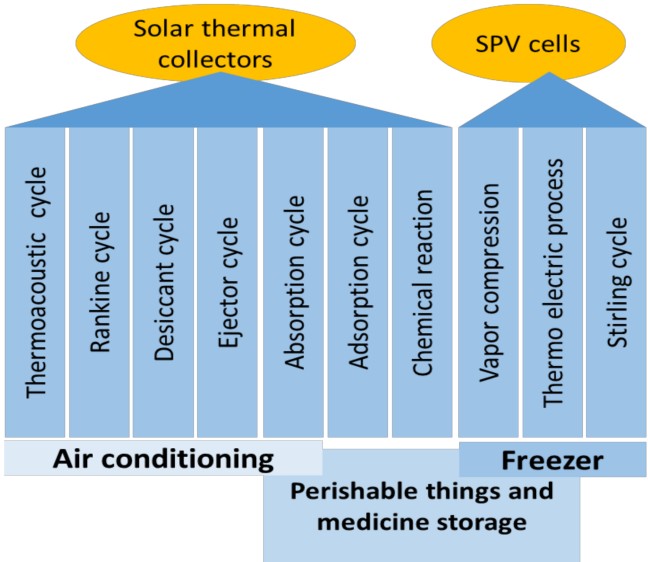

**Figure 3.** Different types of solar cooling systems.

*4.4. Solar Daylighting*

The thermal comfort of building occupants is also affected by the lighting systems installed throughout. The inefficiency of most common lighting technologies, including incandescent bulbs, floodlights, and spotlights, implies that most of the energy they use is wasted as heat rather than light. Therefore, efficiency of the lighting systems—producing less heat output and more light—is the scope in most modern building industries. In addition to this, according to the report by the International Commission on Illumination (CIE), lighting has begun to shift its focus from lighting for visibility purposes to a broader definition of the quality of illumination, which includes human needs, architectural integration, and economic constraints, including energy as well [57]. Typically, lighting accounts for 15–20% of the global energy consumption in the building sector [58]. Of this, more than half of the power source for lighting is coming from fossil fuels around the globe. Hence, adequate lighting systems are needed inside the built environment. Certain studies have been conducted to evaluate the performance of lighting systems, demonstrating that some issues should more commonly be taken into consideration for a healthy and comfortable life of the occupants.

Radiation from lighting, including ultraviolet, visible, and infrared radiation, may cause damage to the eye and skin of the occupants of buildings [59]. Artificial lighting should be regarded as a significant contributor to SBSs in general, and there are practical techniques of enhancing lighting conditions and avoiding harmful effects in lighting of buildings. To achieve a harmonious relationship between lighting and architecture, it is essential to keep in mind the aesthetic, functional, and energy-efficient features of architectural lighting [60]. Aesthetic refers to when designers and architects consider the emotional influence of the lighting and architectural balance on the occupants. The lighting must seem a specific manner, but it must also fulfil its primary goal of illumination. In order for residents to feel comfortable while traversing a building, some areas should be well-lit. Energy efficiency may be achieved by ensuring that the vast majority of light reaches its intended destination and that less light is wasted. The building will be more efficient if the quantity of wasted light is decreased.

Compared to traditional lighting systems, solar daylight is the ideal alternative for different types of buildings because of its abundant potential. Windows and skylights give natural light and ventilation, and the link to the outside provides insight into day/night cycles, weather, and recreational opportunities, along with the thermal comfort enhancement. There are huge energy savings to be had by switching from using electric lights and air conditioning to using natural light for all illumination needs. While there are many benefits to incorporating daylighting into commercial buildings, the implementation process may be difficult [61,62]. These energy savings can only be realized with the help of lighting controls that modify the amount of artificial light, produced by electricity, depending on the amount of natural light present. Skylights are an excellent and cost-effective way to bring natural light into single-story homes and the upper stories of multi-story structures [63]. Skylights let natural light into buildings, which is, hence, their principal use; if their glazing material has specular transmittance, they may let in light and provide a glimpse of the sky [64]. On the other hand, diffuse transmittance glazing is often used in skylights to mitigate the glare of direct sunlight. Diffuse glazing materials obscure views to the outside, but allow for a connection to the weather and the passage of time via observation of the sky and daylight [65].

Along with occupants' thermal comfort improvement with the daylighting systems, there are significant numbers of benefits that have been identified through previous studies. Vision is the most developed sense in humans, and is highly dependent on the availability of appropriate light. Vision performance is affected by lighting conditions. Daylight is an excellent source of illumination for visual performance. It is a flicker-free light source, with a constant distribution of spectral power over the whole visible spectrum [66]. The high luminance enhances visual acuity by facilitating the distinction of minute details. The spectral power distribution of daylight provides ideal colour rendering and permits greater colour discrimination than the majority of artificial lighting. However, the directionality of both daylight and artificial lighting may generate shadows that improve the intricacies of three-dimensional operations [67].

The lack of exposure to sunshine seems to be associated with the development of myopia or short-sightedness [68]. Thus, exposure to sunshine at substantially greater levels than those generally encountered inside may be crucial for avoiding myopia. The complicated protective impact of sunshine may rely on several interdependent factors, such as duration and timing of exposure, wavelength, and intensity. The spectral component of sunshine exposure may influence the visual performance of colour perception. Prior research indicates that colour deficits are more prevalent at northern latitudes when darkness covers a larger portion of the day than near the equator, where they are quite infrequent [69]. A study of visual perception in individuals born below and above the Arctic Circle in various seasons revealed that a reduction in daylight and an increase in exposure to twilight and electric lighting during infancy altered colour sensitivity; participants born in autumn above the Arctic Circle demonstrated the lowest overall colour performance [69]. Appropriately-timed illumination may entrain the circadian system, which is essential to favourably influence sleep quality, health, mood, and cognitive capacities [70]. Due to temporal fluctuations in spectral power distribution and intensity, daylight is the natural time signal for synchronizing the circadian clock and the sleep–wake cycle [71].

The circadian system receives light input from photosensitive retinal ganglion cells, which are especially sensitive to the short-wavelength blue component of light [72]. These cells are linked to the circadian clock and other regions of the brain, influencing nonvisual activities predominantly. The combination of daytime exposure to strong, short-wavelength light and nocturnal avoidance of light is vital for circadian function [73]. Research conducted in the Antarctic area reveals that the sleep quality of base employees is superior during the season of the year with daylight, with its prevalently greater light levels, as contrasted to the sleep quality during the polar winter, with only artificial illumination [74]. In terms of maintaining healthy sleep–wake cycles, blue-enriched light is more effective than intense white light [75]. Less exposure to sunshine during the day and exposure to

artificial light after sunset might delay the circadian clock in indoor environments, resulting in difficulty falling asleep at night and rising on time. Extensive exposure to natural light synchronizes the circadian system with the solar day.

Another study advocated treating and preventing circadian misalignment by increasing daytime illumination and decreasing night time illumination [76]. This necessitates using dynamic variations in spectral composition and using architectural solutions to increase the amount of daylight entering buildings. Simulating the high amplitude temporal dynamics of daylight with electric illumination demands a substantial amount of energy. Daylight is thought to be the optimal and most suitable light source for circadian entrainment. The strongest zeitgebers are dawn and dusk signals, which do not need intense light, but rather a pattern of daily variation with sunrise and sunset. They are dependent on the season and latitude [77]. Compared to static lighting, dynamic lighting simulating a natural sunrise through a change in colour temperature from 1090 K to 2750 K and illuminance at the eye (0–250 lx) resulted in improved subjective mood and well-being, enhanced cognitive performance, and demonstrated that it could be a potential cardiac vulnerability protector during the critical morning hours [78].

Dynamic lighting that featured lower lighting conditions and colour temperatures in the mornings and nights resulted in much more melatonin synthesis one hour before sleep than static lighting [79]. Daylight outside gives inherent temporal dynamics. Therefore, going outdoors is the easiest method for obtaining sufficient circadian input. In buildings, the shape and facade, as well as the choice of window-glazing material and shading system, affect the intensity, colour, and distribution of interior daylight. Daylighting conditions for a building's occupants also rely on their distance from the window, the shape of the room, and the surface reflectance. Depending on the daylighting design, indoor daylight may often offer a sufficient stimulation and support for the circadian system, allowing it to continue to serve as the standard light source for circadian support.

*4.5. Electricity from Solar Energy*

The majority of traditional occupants' thermal comfort improvement applications, such as HVAC, need power to operate. Nevertheless, as stated in the preceding sections, the majority of global electricity comes from fossil fuels, and their negative effects are quite severe, resulting in highly severe occurrences inside and outside of the buildings. Following the same applications and utilizing the same equipment but changing the source of power production to solar energy will be the best solution. The utilization of conventional electricity sources, such as those derived from fossil fuels, can result in a multitude of adverse effects on the environment. These effects include the release of harmful greenhouse gas emissions, which can indirectly contribute to the destruction of indoor environmental quality and exacerbate the issue of global warming. Despite this, electricity remains a crucial component in enhancing the comfort of occupants. Solar energy is a sustainable and environmentally-sound energy source that produces no harmful emission during electricity generation. Its utilization is conducive to improving indoor environmental quality and promoting greater environmental friendliness. This will assist in reducing the harmful effects of these energy sources. The prevalent approximation for the mean duration of paying back the investment for solar panels is between six and ten years. The variability in the duration of time required to pay off solar panels and the corresponding monthly savings can be attributed to a multitude of factors. The estimated energy payback periods for rooftop photovoltaic (PV) systems are 4, 3, 2, and 1 years for multicrystalline-silicon, thin-film modules, anticipated multicrystalline, and anticipated thin-film modules, respectively, based on energy paybacks ranging from 1 to 4 years and estimated life expectancies of 30 years [80].

Facades (Figure 4), roofs, and sunshades are the three main inclusion points for solar photovoltaics (SPVs) in building construction as building-integrated photovoltaic systems (BIPV) (Figure 5). Curtain walls, spandrel panels, and glass all make up several kinds of facade systems [81]. The most cost-effective and efficient way to install renewable energy

systems in buildings is via BIPV, especially in dense metropolitan regions where land for development is expensive and limited. Optimizing energy efficiency within the building's energy consumption is the first stage in every BIPV implementation. This holistic method makes the most of energy savings and BIPV systems by combining energy conservation, energy efficiency, building envelope design, and PV technology.

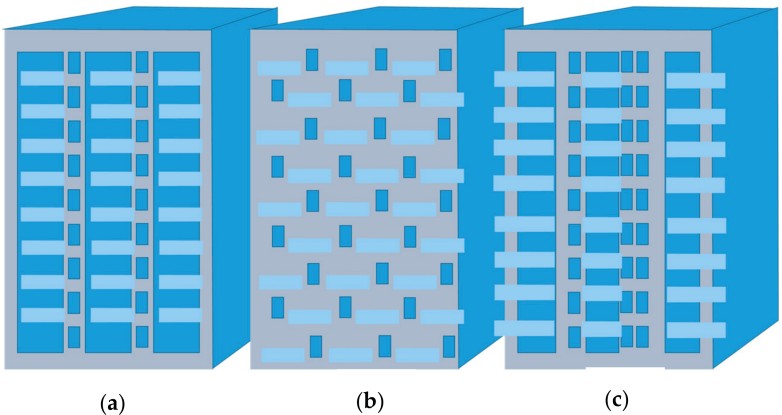

(a)                                        (b)                                        (c)

**Figure 4.** Various types of facade arrangement in high-rise buildings: (**a**) aligned balconies, (**b**) staggered balconies, (**c**) side balconies.

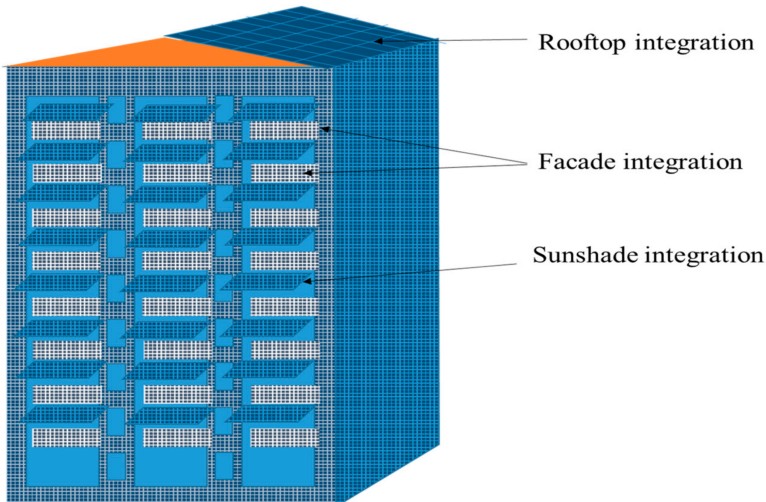

**Figure 5.** Different types of high-rise building solar photovoltaic integrations.

Figure 4 shows several different facade arrangements that can be utilized to install solar photovoltaic modules, including (a) a straight-across construction for the balconies, (b) a staggered structure, and (c) a side-by-side structure. Figure 5 depicts the potential integration of solar photovoltaic modules in a high-rise structure, including integration of facades, roof, and sunshades.

Design, shading, and homeowner preferences are the three main limitations of BIPV systems [82]. The design of the building must take into account how simple it is to install and incorporate, as well as adhere to all applicable codes. Due to the high voltage involved, its installation is often more difficult and needs the assistance of professionals trained in PV system technology. There may be more complications with regards to safety and water tightness with such a layout. There is also the issue of how a building is oriented, which might restrict the amount of sunlight that can reach a PV system. In the northern hemisphere, facing south maximizes radiation across the PV module surface but also produces a lot of heat. The BIPV panels have to work at greater temperatures since they are always exposed to the sun and there is not enough airflow to cool them. The process

of adapting an old building is complicated by a wide variety of extra limitations and unknown expenses. Recladding a building with a PV curtain wall, for instance, is quite similar to recladding with a normal curtain wall, with the exception of accommodating wiring. Even while retrofit applications may function as well as brand-new installations, it is important to assess each refit project on its own merits. Planning PV panels during the construction phase of a building is usually the most efficient method, but designing a multi-story building either way involves more work and expense.

Many alternative strategies are used to switch out traditional window panes with PV panels in buildings. Any PV panel may be used in replace of spandrel glass as long as its dimensions and aesthetic qualities are appropriate for the building's design. Thin-film amorphous PV modules are, aesthetically, extremely-near matches; they are also very thin and lightweight, but their efficiency is poor [83]. PV windows with just a partial transparency cannot be used as a substitute for regular glass. Still, PV panels are a great alternative to semi-transparent windows in a wide variety of settings. High-rise buildings' outside curtain walls sometimes have extensively tinted or patterned glass to reduce heat input and regulate glare. Since most large-area thin-film modules have thin lines scribed through the cell material, they are somewhat transparent.

In either monocrystalline or polycrystalline BIPV modules, the light travels via the interstices between the PV cells, and the amount of light let through may be adjusted with relative ease. Still, similar to conventional glass, PV cells need a little tweaking to meet all of the design criteria. Since metal substrate and superstrate PV panels are so adaptable, they may be used in lieu of sheet metal anywhere on a building's roof or exterior [84]. However, they are not as practical as window-based systems, due to the high expense of tailoring the system to each individual structure. Thus, thin-film modules with metal substrates find widespread deployment in roofing systems. Because of this, they are less useful on building structures with limited roof space. It would seem that tailoring PV panels made of metal to the needs of a high-rise building structure is more challenging and expensive than tailoring PV panels made of metal to the needs of an integrated metal-based single-story building. Several typical practices exist for installing PV panels within a structure. All of these layouts include installing PV panels on top of or incorporating them into an existing roof.

Angled glass windows, sunrooms, atriums, greenhouses, and slanted walls all fall under this category of slanted or sloping roof and wall surfaces. Aluminium-framed, tinted, laminated glass or plastic-glazing modules are examples of these semi-transparent glazing systems. They are a common component of buildings' natural lighting systems. Because they are the same size and form as tinted, laminated glass pieces, many commercially available PV modules may be used for these purposes. When it comes to office buildings, diffuse daylighting is preferred, since direct sunlight may create overheating and glare [85]. While the construction parameters of a vertical PV panel assembly with or without windows are identical to those of a sloped-glazing assembly, the vertical orientation of the panels results in a lower PV output. In contrast, curtain walls may accommodate a greater variety of PV products. Non-transparent modules may also be used in places such as curtain walls on tall structures. It is possible that some of the curtain wall glazing might be made out of PV panels that are only partially transparent and have a middling optical quality [86]. On the other hand, there are certain trade-offs to be made in this layout, such as with glare, overheating, and the number of PV panels installed on the facade.

Having PV panels installed on slanted walls increases efficiency because of the angle at which the panels are positioned, but makes the building harder to design. Additionally, it has the potential to induce some self-shading. However, unlike window-based technology, this design is not limited to using just those kinds of PV panels now available on the market. When sunshades are permanently installed, they block out more light, but provide better shading, which is great for reducing glare. Because of this, tilting the PV panels might increase their efficiency [87]. Furthermore, shade components are well-suited to house PV laminates, since they face the sun, often have a flat surface, and provide back ventilation which helps to disperse the produced heat. Most often, a metal frame has been used to

attach ordinary PV modules to the outside of the structure. Furthermore, construction is simplified, since the process is identical to that of erecting standard sunshades. On the other hand, it is to be anticipated that some of the PV panels will be shaded by neighbouring structures or by other PV panels. Any kind of PV cells, including traditional PV panels, are usable, the same as inclined PV panels. Designers have greater leeway with this choice. This kind of construction makes upkeep less of a hassle.

## 5. Discussion

Despite the widespread recognition of the long history and relevance of thermal comfort enhancement applications in the built environment, no agreement has been reached on how to best integrate solar energy into these applications so as to maximize thermal comfort for a wide range of building occupants in a way that is also sustainable, cost-effective, and harmless to human health. Perceived accessibility is a concept that has not been studied extensively until recently, despite its inception in the middle of the twentieth century. Even though, many obstacles stand in the way of deployment of solar energy utilization techniques in building, preventing it from successfully competing with conventional energy sources [88,89].

The social, economic, technical, and regulatory contexts are the usual perspectives through which to examine and understand barriers. The transition from fossil fuels to solar energy usage in the built environment has been met with public criticism and opposition due to worries about the undesired stress on the solar power systems and components [90,91]. The foundation of sustainable development is the satisfaction of occupants' or users' need through generally-accepted technological systems and appropriate policy and regulatory instruments, but there are a number of factors that limit the spread of knowledge and understanding among the general public.

The lack of familiarity with RET (renewable energy technology) and reservations of building owners or shareholders regarding the financial sustainability of RE installation are two of the main obstacles to public understanding [92]. While there is widespread support for renewable energy in principle, local opposition to its implementation is a significant barrier to widespread adoption that can only be overcome through effective education and outreach [93]. Building owners may encounter pushback from neighbours, city authorities, community organizations, special interest groups, and even environmentalists when they propose renewable energy initiatives. Public opposition stems from a number of sources, including concerns about the environment, the possibility for visual harm, and a lack of community involvement [94,95].

High initial investment, a dearth of financial institutions, an absence of investors, competition from fossil fuels, and a lack of subsidies in comparison to conventional fuel all contribute to economic and financial hurdles [96,97]. Some countries' renewable energy subsidies are much lower than their conventional energy subsidies. Thus, renewable energy remains at a competitive disadvantage. Government subsidies for fossil fuel power plants are hindering the widespread adoption of cleaner, more efficient alternatives. Financial for renewable energy projects at rates as low as those made available for fossil fuel energy projects is very difficult to come by due to a lack of financing institutions [98].

There is a dearth of resources for funding renewable energy projects. This reflects the fact that the investments are seen as somewhat hazardous, which discourages potential investors. There is a significant lag time between the initial investment and the return for renewable energy projects due to the lower efficiency of renewable technologies [99]. Users and producers alike may have just a small amount of cash on hand, but both need to invest in a plant in order to render it operational. Even when money is available for conventional energy projects, tight lending regulations make them difficult to implement. High payback periods, combined with the high cost of capital and the frequent absence of money, render many ventures unfeasible. Fuel prices now cover the costs of exploration, production, distribution, and consumption in practically all nations, but they do not account for the harm done to the environment and society as a result of using these fuels [100]. The price

of old fuels often excludes the hidden costs, such as their negative impact on health and the environment. Knowing the effects of using fossil fuels for energy production is crucial for determining the true cost of doing so.

The widespread implementation of solar energy systems in high-rise buildings faces a number of legitimate technological barriers, including a lack of infrastructure, insufficient knowledge of operations and maintenance, insufficient research and development initiatives, and technical complexities such as energy storage and the absence of standards in some countries [101]. One obstacle hindering the widespread adoption of renewable energy is the lack of access to the cutting-edge technology necessary to harness their full potential. This is particularly true in developing nations. The price tag for acquiring such devices is prohibitive even if the technology itself is readily accessible [102]. There is a dearth of understanding regarding operation and maintenance of solar energy systems since the technology is still relatively young and not fully matured. If a plant is not being run at its full potential and routine maintenance is not being performed, efficiency will never be realized.

Equipment, components, and replacement parts that are not locally available will drive up manufacturing costs significantly, since they will have to be imported from other nations, where they'll likely be more expensive. Inadequate funding for research and development (R&D) prevents solar energy from becoming economically competitive with fossil fuels. Due to the considerable risks associated with solar energy's early stage of development, governments and energy companies are hesitant to invest in research and development [103]. Storage of energy is a key technological challenge for the solar energy industry today [104,105]. Electricity networks cannot function unless supply and demand are balanced, yet the sun's supply is not constant, despite its unlimited quantity. Large batteries that can make up for when a renewable resource is unavailable are needed to fix this problem.

Insufficient national policies, bureaucratic and administrative impediments, inadequate incentives, unworkable government objectives, and a lack of standards and certifications have all worked against solar energy's rapid expansion [106]. In addition to being necessary for a country's sustainable growth, energy sector regulatory rules that address the discrepancy between renewable and non-renewable sources of power are also necessary [96]. Due to a lack of clear policies, the execution of the subsidies is a source of contention between the numerous government agencies involved. The development phase of the project takes longer than necessary because of a lack of coordination between different authorities and lengthy lead times in obtaining authorization [107].

Permission is costly because of lobbying efforts. All of these things make it harder to motivated investment in solar energy, and lengthen the time it takes to launch a project. To better promote these technologies to the building owners as stable and growth-oriented, policies should be formulated to provide clear insight into key legislative and regulatory challenges. Responding and reacting more quickly may help governments address this discrepancy. In order to guarantee that the machinery and components created or acquired from abroad meet the requirements of the importing firm, standards and certifications must be provided [96]. These approvals ensure that the facility is being run in accordance with all applicable regulations.

## 6. Future Scope

Solar energy deployment is critical for resolving both energy and climate change issues. However, the growth and penetration of solar energy globally among building sectors are hampered by the constraints (social, economic, technical, and regulatory) existent in this sector. Investors and businesses are discouraged from joining and investing in renewable energy because of the excessive red tape involved in the implementation of solar power. Investment in the energy industry is being discouraged due to a lack of consistency between federal and provincial laws.

The penetration of renewable energy sources is lower in nations with very complex administrative processes than in those with simple and easy procedures. Energy is essential to the growth of every nation's economy and society. Energy shortages may be alleviated without causing irreparable harm to the environment if more countries adopt the use of renewable energy sources. Inevitably, this shift would lead to a rise in demand for personnel skilled in the planning, construction, operation, and upkeep of renewable energy projects. Increased market penetration leads to economies of scale, which in turn lowers per-unit production costs and, ultimately, the price paid by consumers.

Greater investment in solar energy projects will result from the increased confidence of building owners. The greater the availability of green energy, the greater the potential advantages, since there will be fewer negative effects on the environment. This may aid in the preservation of the planet's ecology. The development of more affordable methods to produce, store, and disperse solar energy is urgently needed. However, compared to conventional energy, insufficient research and development initiatives are undertaken. This is because there is little hope for these kinds of projects, since building owners cannot expect to profit.

## 7. Conclusions

The present study analyzed and compared different novel approaches for utilising solar power and leveraging its advantages to improve the thermal comfort of occupants both indoors and outdoors. The study identified several alternative strategies for buildings to achieve passive ventilation, solar cooling, solar heating, solar power generation, and solar daylighting as compared to conventional practices. Additionally, the article discusses the adverse health and environmental impacts resulting from conventional practices, including HVAC, lighting systems, and other applications to improve thermal comfort. This article highlights the health and environmental advantages of implementing the aforementioned techniques in buildings, in addition to enhancing the thermal comfort of the occupants. It is possible to substantially decrease a building's dependence on non-renewable energy sources for its energy requirements, thereby mitigating the building's environmental impact and reducing its inhabitants' energy demands. Nonetheless, numerous challenges exist, and surmounting them would enhance the probability of global implementation and contribute to the realization of a world free from energy-related stress in the foreseeable future. The aforementioned statement holds true at both the micro and macro levels. This is achieved by implementing efficient protocols and policies, devising innovative methods to tackle real-world predicaments, and sharing findings from relevant research and advancement. The learnings of the study are summarized and pointed out in these lessons learned:

- A wide range of building-related technologies exist to improve occupants' thermal comfort, from space and water heating to ventilation, air conditioning, and lighting;
- Energy requirements for operating thermal comfort enhancement technologies are a major obstacle for the building sector globally;
- In spite of the fact that fossil fuels provide the majority of the required energy, their utilization and the strategies that have traditionally been used to put that energy to use have been linked to a number of detrimental outcomes for the environment and human health;
- Renewable energy is one viable solution to the problem of meeting the globe's growing energy needs, and cutting-edge, novel approaches are being sought to mitigate the problems caused by the widespread use of conventional methods for improving thermal comfort;
- There are many ways in which solar energy may be utilized to improve the thermal comfort of the built environment, and as solar power becomes more widely-accessible throughout the globe, it will also become a more reliable and sustainable energy source;
- The most efficient use of conventional energy sources and conventional thermal comfort-enhancing applications may be attained via the best techniques of incorporating solar energy harvesting setups into the structure.

**Author Contributions:** Conceptualization, A.M.M.I. and E.J.; methodology, A.M.M.I. and E.J.; software, H.-W.C. and B.R.; validation, E.J.; formal analysis, A.M.M.I.; investigation, H.-W.C. and E.J.; resources, B.R.; data curation, A.M.M.I. and E.J.; writing—original draft preparation, A.M.M.I.; writing—review and editing, E.J. and H.-W.C.; visualization, B.R.; supervision, E.J. and H.-W.C.; project administration, A.M.M.I.; funding acquisition, H.-W.C. All authors have read and agreed to the published version of the manuscript.

**Funding:** This research received no external funding.

**Institutional Review Board Statement:** Not applicable.

**Informed Consent Statement:** Not applicable.

**Data Availability Statement:** Not applicable.

**Conflicts of Interest:** The authors declare no conflict of interest.

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
