# Peer review of "Enhancing Occupants’ Thermal Comfort in Buildings by Applying Solar-Powered Techniques"

_2673-8945, doi:10.3390/architecture3020013_

Round 1

Reviewer 1 Report

I have reviewed this article about thermal comfort of the occupants

  It seems to me that the topic has been insufficiently studied in the article. This manuscript works as a kind of review article or the first chapter of a handbook on solar-powered solutions for architecture. The methodology of the article is not focused on real examples it just describes a repertoire of well wishes without order or concretion.

 The authors could perhaps improve their methodology by taking into account real examples of building design and planning.

 The authors should be more careful limiting their research to a few selected cases which they analyse in more detail with simulations and monitoring.

 The article in summary seems disjointed and bland and it surely needs of more research, I miss some more opportunities to enhance the knowledge on the matter.

 The conclusions are not consistent, they are limited to intentions and seem a bit week.  The results should be much more straightforward in order to publish the article.

Summary of evaluation: This article could be interesting but it shows several drawbacks from the scientific point of view. I hint to a thorough improvement of the manuscript to increase the transmission of knowledge.

I have reviewed this article about thermal comfort of the occupants

  It seems to me that the topic has been insufficiently studied in the article. This manuscript works as a kind of review article or the first chapter of a handbook on solar-powered solutions for architecture. The methodology of the article is not focused on real examples it just describes a repertoire of well wishes without order or concretion.

 The authors could perhaps improve their methodology by taking into account real examples of building design and planning.

 The authors should be more careful limiting their research to a few selected cases which they analyse in more detail with simulations and monitoring.

 The article in summary seems disjointed and bland and it surely needs of more research, I miss some more opportunities to enhance the knowledge on the matter.

 The conclusions are not consistent, they are limited to intentions and seem a bit week.  The results should be much more straightforward in order to publish the article.

Summary of evaluation: This article could be interesting but it shows several drawbacks from the scientific point of view. I hint to a thorough improvement of the manuscript to increase the transmission of knowledge.

Author Response

Respected reviewer,

We express our gratitude for the valuable reviews and insightful remarks provided on our manuscript. Please see the attachment for our responses.

Reviewer 2 Report

The authors develop a research whose purpose is to improve the thermal comfort of building occupants by applying solar energy techniques.

The article is pertinent and relevant; also very appropriate for the magazine. I have some comments that will help improve the quality of the article.

1. How do the authors simulate the contributions of

natural lighting, passive ventilation, thermal applications, cooling applications and power generation. Better justify these values in the methodology.

2. Enrich the state of the art with more impact references such as:

-Berouine, A., Ouladsine, R., Bakhouya, M., & Essaaidi, M. (2022). A predictive control approach for thermal energy management in buildings. Energy Reports, 8, 9127–9141. https://doi.org/10.1016/j.egyr.2022.07.037

-Roccotelli, M., Rinaldi, A., Fanti, M. P., & Iannone, F. (2021). Article building energy management for passive cooling based on stochastic occupants behavior evaluation. Energies, 14(1). https://doi.org/10.3390/en14010138

-Fernandez-Antolin, M. M., del-Río, J. M., & Gonzalez-Lezcano, R. A. (2019). Influence of solar reflectance and renewable energies on residential heating and cooling demand in sustainable architecture: A case study in different climate zones in Spain considering their urban contexts. Sustainability (Switzerland), 11(23). https://doi.org/10.3390/su11236782

3. It would be interesting to tabulate or summarize the effects on health of the different situations of lack of thermal or hygrothermal comfort, or of lighting or acoustics that can affect people.

4. How are the habits of the occupants simulated?

5. What type of infiltrations is considered for the energy simulations.

6. It would be interesting to explain the energy contribution or saving due to all the passive strategies considered.

7. What percentages of savings can be achieved through the generation of solar energy through solar modules and how long the capital investment will be amortized.

8. In what way could the quality of the indoor environment be guaranteed; with heating, ventilation and air conditioning systems powered by solar energy

Author Response

(The authors gave the same response as above.)

Round 2

Reviewer 1 Report

The authors have sufficiently improved the article

Reviewer 2 Report

The authors have answered all the questions put to them. The article has improved considerably.

I have nothing further to add